# Detegmentation of Organs-at-Risk for Head and Neck Cancer

**Shan Wang**[1]                                                            202300172023@mail.sdu.edu.cn
**Jax Luo**[2]                                                                          jluo5@mgh.harvard.edu
**Xu Qiao**[1]                                                                          qiaoxu@sdu.edu.cn
**Nazim Haouchine**[3]                                               nhaouchine@bwh.harvard.edu
**Scott Raymond**[2]                                                               raymons3@ccf.org

[1] *School of Control Science and Engineering, Shandong University, Shandong, China*

[2] *Neurological Institute, Cleveland Clinic, OH, USA*

[3] *Brigham and Women's Hospital, Harvard Medical School, MA, USA*

**Editors:** Accepted for publication at MIDL 2025

## Abstract

Segmentation of organs-at-risk (OARs) is a critical step in radiation therapy planning for head and neck cancer (HNC). Recent advancements in fine-tuned foundation models have demonstrated the potential to surpass conventional segmentation methods when provided with appropriately sized box prompts. We present a practical Detegmentation framework that integrates a detection network to autonomously generate box prompts for training and testing a foundation model for OARs segmentation in HNC. Our approach outperforms leading methods from the most recent HaN-Seg Challenge, highlighting its strong potential for clinical application.

**Keywords:** Detegmentation, Detection, Segmentation, Foundational Model, Head and Neck Cancer

## 1. Introduction

Head and neck (HNC) cancer ranks among the most prevalent malignancies worldwide. In radiation therapy for HNC, accurate segmentation of organs at risk (OARs) is critical to minimizing post-treatment radiation-induced complications (Ye et al., 2022). Recent advancements in fine-tuned foundation models have demonstrated the potential to surpass conventional segmentation methods when provided with appropriately sized box prompts—rectangular regions that indicate the location of the target object (Kirillov et al., 2023; Xu et al., 2024, 2025; Fan et al., 2019; Chen et al., 2024). However, manually delineating these boxes for OARs is impractical, as it requires substantial clinician input and is susceptible to intra- and inter-rater variability. In this study, we propose a Detegmentation framework that integrates a detection network to autonomously generate fitted box prompts for each target OAR, enabling streamlined training and testing of a foundation segmentation model without clinician intervention. Our approach outperforms leading methods from the most recent HaN-Seg Challenge(Podobnik et al., 2023), highlighting its strong potential for clinical application.

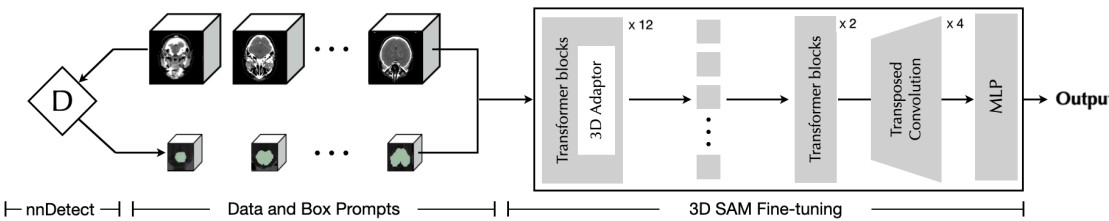

Figure 1: An overview of the proposed Detegmentation framework. $D$ represents the incorporated detection model.

## 2. Methods

The Detegmentation framework comprises an nnDetect(Baumgartner et al., 2021) model for object detection and a 3D adaptation of Segment Anything Model (SAM)(Chen et al., 2024), which can be fine-tuned(Hu et al., 2021) for specific OARs segmentation. As illustrated in Fig. 1, we first train the nnDetect model $D$ to automatically predict the locations and bounding box coordinates of the target OARs. These coordinates are then used as box prompts to fine-tune the 3D SAM for enhanced OAR segmentation. During inference, the test case is first processed by $D$ to generate the corresponding box prompt. This prompt, together with the input image, is then fed into the segmentation model to produce the predicted segmentation label.

## 3. Experiments

We assessed the proposed framework using the brainstem and left parotid gland structures from the publicly available HaN-Seg challenge dataset(Podobnik et al., 2023), which was partitioned into 30 cases for training and 12 cases for testing.

Our method demonstrates reliable detection performance, achieving a sensitivity of 1.0 for both the brainstem and the left parotid gland. The Dice similarity coefficient between the ground truth and predicted bounding boxes is 0.78. To ensure complete coverage of the target structures during segmentation, we expanded the input bounding box by 5 voxels in each direction.

As summarized in Table.1, the proposed Detegmentation framework yields a mean Dice score of 0.895 and an HD95 of 1.8 mm for brainstem segmentation, and a Dice score of 0.886 with an HD95 of 2.0 mm for left parotid gland segmentation, surpassing the top performing nnU-Net(Isensee et al., 2018) and DynUNet(Yang et al., 2025) methods in the latest HaN-seg challenge. Fig. 2 presents qualitative results for the segmented brainstem and left parotid gland.

## 4. Discussion and Conclusion

We present a practical Detegmentation framework that integrates a detection network to autonomously generate box prompts for training and testing a foundation model for Organs at Risk segmentation in Head and Neck cancer. Our method enables accurate OAR

Table 1: Quantitave segmentation results comparing the proposed Detegmentation framework with leading methods in the HaN-seg challenge

| OAR | Mean | nnU-Net | DynUNet | Ours |
|---|---|---|---|---|
| Brainstem | DSC | 0.885 | 0.849 | **0.895** |
| | HD95 | 3.9 | 4.7 | **1.8** |
| Parotid gland(L) | DSC | 0.867 | 0.851 | **0.886** |
| | HD95 | 5.1 | 5.1 | **2.0** |

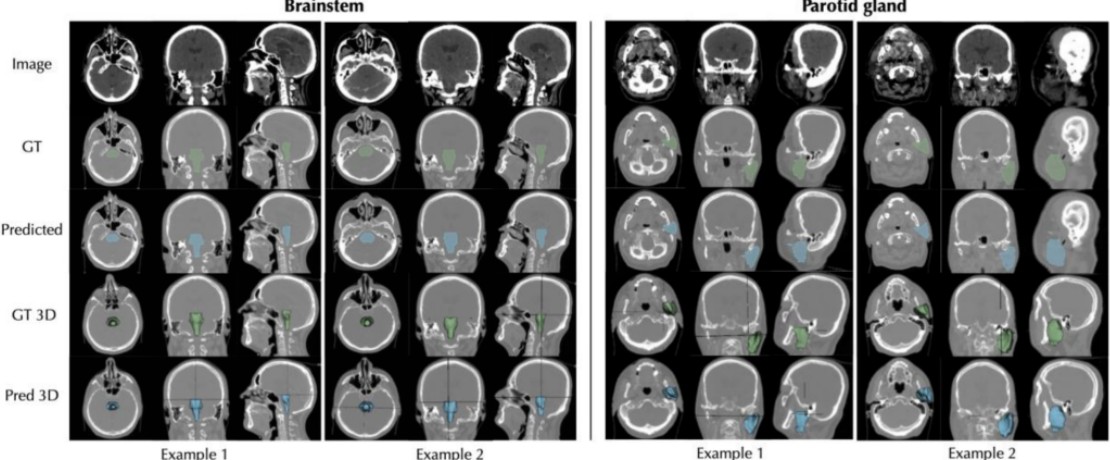

Figure 2: Qualitative results showcasing the predicted segmented brainstem and the left parotid gland labels, along with their 3D renderings. GT denotes the ground truth label.

segmentation without clinician intervention, highlighting its strong potential for clinical application.

Future research will aim to extend the proposed framework to encompass a broader range of OARs, with particular emphasis on smaller and more challenging structures such as the optic chiasm and optic nerves. Additionally, more advanced object detection models will be explored to address instances where the current nnDetect model underperforms. Another important direction involves integrating multiple imaging modalities (e.g., CT and PET) as input, with the objective of further enhancing segmentation accuracy.

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
