# OpenReview forum: "Detegmentation of Organs-at-Risk for Head and Neck Cancer"
_MIDL.io/2025/Short_Papers — MIDL 2025 - Short Papers_

### Official Review · Reviewer_d6FU · 2025-04-29

**Rating:** 3
**Confidence:** 5

**Summary:**

The paper presents a new Detegmentation framework designed for segmenting organs-at-risk (OARs) in head and neck cancer. It combines an automatic detection network (nnDetect) with a 3D-adapted Segment Anything Model (SAM). The method autonomously generates bounding box prompts. The proposed method is evaluated using the HaN-Seg dataset, demonstrating superior segmentation results (Dice scores of 0.895 for brainstem and 0.886 for left parotid gland) compared to established methods like nnU-Net and DynUNet.

**Strengths:**

The approach reduces manual clinician workload by automating bounding box generation.

Achieves state-of-the-art segmentation performance, outperforming current leading methods, such as nnUNet.

**Weaknesses:**

Limited evaluation scope, involving only two OARs (brainstem and left parotid gland); results might not generalize fully across all OARs.

No detailed analysis was provided on failure cases or the robustness of the detection network in more challenging clinical scenarios.

Lacks extensive comparisons with additional competitive contemporary models or methodologies beyond the two cited methods.

---

### Decision · Program_Chairs · 2025-05-01

Accept